# Clinical Features and Genetic Findings of Autosomal Recessive Bestrophinopathy

**DOI:** 10.3390/genes13071197

**Published:** 2022-07-04

**Authors:** Hae Rang Kim, Jinu Han, Yong Joon Kim, Hyun Goo Kang, Suk Ho Byeon, Sung Soo Kim, Christopher Seungkyu Lee

**Affiliations:** 1Department of Ophthalmology, Institute of Vision Research, Yonsei University College of Medicine, Seoul 03722, Korea; khr1412@chamc.co.kr (H.R.K.); kyjcolor@naver.com (Y.J.K.); hgkang08@yuhs.ac (H.G.K.); shbyeon@yuhs.ac (S.H.B.); semekim@yuhs.ac (S.S.K.); 2Department of Ophthalmology, CHA Bundang Medical Center, CHA University College of Medicine, Seongnam 13496, Korea; 3Department of Ophthalmology, Institute of Vision Research, Gangnam Severance Hospital, Yonsei University College of Medicine, Seoul 06273, Korea; jinuhan@yuhs.ac

**Keywords:** autosomal recessive bestrophinopathy (ARB), *BEST1* gene

## Abstract

Autosomal recessive bestrophinopathy (ARB) is a rare subtype of bestrophinopathy caused by biallelic mutations of the *BEST1* gene. ARB is characterized by multifocal subretinal deposits accompanied by macular edema or subretinal fluid, hyperopia, co-existing narrow angle, and a marked decrease in electrooculogram. However, little is known about the genetic variants and specific clinical features of ARB. This is an observational case series of patients with a clinical and genetic diagnosis of ARB who underwent multimodal imaging. We describe ten patients from nine unrelated families with six known variants and three novel missense variants: c.236C→T, p.(Ser79Phe); C.452C→T, p.(Leu151Pro); and c.650C→T, p.(Trp217Met). The most common variant was c.584C→T, p.(Ala195Val), observed in six patients, without correlation to the severity of the phenotype. All patients manifested bilateral multifocal subretinal deposits and subretinal fluid throughout the follow-up period, while intraretinal fluid was found in approximately half of the eyes. The extent or chronicity of the fluid collection did not correlate with visual acuity. Angle-closure glaucoma was present in five eyes. Three patients had a genetically confirmed family history of ARB, and one patient had a clinically suspected family history. This study reveals novel mutations in the *BEST1* gene and adds to the spectrum of clinical presentations of ARB.

## 1. Introduction

Bestrophinopathy is a spectrum of inherited macular degenerations caused by mutations in the *BEST1* gene [1]. *BEST1* is located on chromosome 11q13 [2,3] and encodes bestrophin-1, a 585 amino-acid calcium-activated Cl^−^ channel localized to the basolateral membrane of retinal pigment epithelium (RPE) [4]. Mutation in the *BEST1* gene might result in abnormal functioning of the protein bestrophin-1, an anion channel in the RPE, leading to a variety of retinopathies [5,6]. The most prevalent variant, Best vitelliform macular dystrophy [VMD, also known as Best disease; Online Mendelian Inheritance in Man identifier (OMIM), 153700], is characterized by prominent central macular lesions that undergo consecutive morphologic changes from characteristic ‘egg-yolk’ appearance in the vitelliform stage to vitelliruptive stage, pseudohypopion state, and atrophic stage [7]. Since the first report in 1905 by Friedrich, a wide array of missense mutations in *BEST1* variants have been reported [8]. Other subtypes include adult-onset vitelliform macular dystrophy (OMIM 153840), autosomal dominant vitreoretinochoroidopathy (OMIM 193220), retinitis pigmentosa 50 (OMIM 613194), and autosomal recessive bestrophinopathy (OMIM 611809) [7].

In 2006, Schatz et al. first reported two related patients with multifocal vitelliform dystrophy and compound heterozygous *BEST1* variants. Burgess et al. then denominated the term “autosomal recessive bestrophinopathy” (ARB) as a new *BEST1*-associated phenotype. Unlike missense mutations in autosomal dominant inheritance of VMD, ARB is caused by a homozygous or compound heterozygous *BEST1* mutation with a modifier effect of the first on the second mutation in the latter [9,10]. ARB is characterized by multifocal diffuse subretinal deposits that appear hyperfluorescent on fundus autofluorescence imaging, accompanied by macular edema or subretinal fluid, hyperopia, and co-existing narrow angle. Electrophysiological characteristics include a marked decrease in light rise on electrooculography (EOG) and relatively preserved electroretinography (ERG) parameters unless photoreceptor cells are severely damaged. Although abnormal EOG findings are crucial for the diagnosis of bestrophinopathy, mutation analysis is necessary to confirm the diagnosis of a specific subtype of bestrophinopathy.

The prevalence of ARB is estimated to be 1/1,000,000 [9]. In Korea, only two patients with ARB from a single family were reported in 2015 [11]. Herein, we report on ten patients with ARB due to mutations in *BEST1*, characterizing their clinical features and genetic mutations.

## 2. Materials and Methods

### 2.1. Study Design and Subjects

In this retrospective cohort study, we reviewed medical records of patients with clinical features of ARB who were followed up at the Department of Ophthalmology of two tertiary referral hospitals in Korea (Severance Eye Hospital and Gangnam Severance Hospital, Yonsei University Medical Center, Seoul) between November 2009 and December 2021. Typical clinical characteristics of ARB include bilateral multifocal vitelliform lesions that appear hyperfluorescent on autofluorescence imaging with or without intraretinal fluid (IRF) or subretinal fluid (SRF). The final diagnosis of ARB was confirmed using genomic data generated by next-generation sequencing (NGS). We excluded three patients without genetic information and one patient with poor image quality who was lost to follow-up.

The study was conducted in accordance with the Declaration of Helsinki and the protocol was approved by the Institutional Review Board (IRB) of Yonsei University Medical Center (IRB approval number: 2022-1285-001). The requirement for informed consent was waived because this study used only anonymized data before the analysis.

### 2.2. Data Collection

Data regarding demographic characteristics (age and sex), duration of symptoms (interval between the reported onset of visual symptoms and diagnosis of ARB), and family history of ocular disease were collected for each patient. For each affected eye, the best-corrected visual acuity (BCVA), refraction, intraocular pressure, slit-lamp and dilated fundoscopy results, presence of ocular comorbidities, and previous treatments before the initial visit were recorded. Medical therapy included intravitreal anti-vascular endothelial growth factor (anti-VEGF) injections and laser photocoagulation treatment.

### 2.3. Ophthalmologic Image Interpretation

Ophthalmologic paraclinical examination findings were assessed by two retinal specialists (H.R.K. and Y.J.K.). Multimodal imaging, including color fundus photography, widefield retinal imaging, spectral-domain optical coherence tomography (OCT; Heidelberg Engineering, Heidelberg, Germany), autofluorescence (AF) imaging at the initial presentation and the latest follow-up visit were reviewed. ERG and EOG data were collected if available.

Vitelliform lesions were defined as well-demarcated yellow subretinal deposits confirmed by color fundus photography and OCT, which appeared hyperautofluorescent on AF images. The presence of IRF (defined as >3 adjacent intraretinal hyporeflective spaces visible on OCT), SRF, and pigment epithelium detachment was evaluated. The extent of SRF was categorized as diffuse if the fluid involved the entire line scan on the OCT B-scan, subfoveal if limited to the juxtafoveal area, or absent. Central retinal thickness (CRT) and subfoveal choroidal thickness (CT) from the central 1-mm subfield were determined manually using the built-in caliper of the Heidelberg software. Outer retinal layer thickening, a thicker layer of the interdigitation zone between the RPE and the ellipsoid zone interface, was also examined. Finally, focal choroidal excavation (FCE) was defined as an area of concavity in the choroid without scleral ectasia or posterior staphyloma detected on OCT [12]. Where available, widefield retinal images were graded for the presence of peripheral drusen-like material, defined as the accumulation of subretinal deposits without a decreased AF signal and the presence of RPE atrophy.

### 2.4. Genetic Analysis

For patients with clinical features of ARB, informed consent for genetic analysis was obtained before DNA testing. For the customized NGS panel, genomic DNA samples extracted from peripheral blood leukocytes according to established protocols were evaluated for causative genes of ARB based on literature reviews, RetNet database (http://sph.uth.edu/Retnet/ accessed on 4 June 2022), and OMIM database (http://www.ncbi.nlm.nih.gov/omim accessed on 4 June 2022). Target enrichment was performed using a customized target enrichment kit (Celemics Inc., Seoul, Korea). Sequencing and bioinformatics analyses were performed as previously described [13]. Briefly, the pooled libraries were sequenced using a NextSeq 550 sequencer (Illumina, San Diego, CA, USA) and the NextSeq Reagent Kit, version 2 (300 cycles). A final diagnosis of ARB was made when compound heterozygous variants in the *BEST1* gene were confirmed, only including variants with “pathologic”, “likely pathologic”, or “uncertain significance” qualifiers among the five-tier classification system. Patients with inconclusive genetic data were excluded from the study, even those with typical ARB phenotypes.

## 3. Results

Ten patients from nine unrelated families were included in the study. Patient characteristics are summarized in Table 1. The mean age at onset of subjective symptoms and at diagnosis was 25.8 ± 13.4 (range, 6–50) years and 34.8 ± 18.8 (range, 6–67) years, respectively. Eight (80%) patients complained of reduced central vision as the initial symptom, while two (20%) patients (patients 4 and 10) were asymptomatic. At the final visit, three patients (patients 6, 8, and 10) subjectively reported gradual decline in vision while others claimed no vision change. The median initial BCVA of the right and left eye were 0.47 (range, 0.025–1.0) and 0.52 (range, 0.025–1.0), respectively. The median final BCVA of the right eye and left eye were 0.39 (range, 0.04–1.0) and 0.42 (range, 0.04–1.0), respectively. There was no significant difference between initial and final BCVA in both eyes (*p* = 0.604 for the right eye; *p* = 0.630 for the left eye). One (10%) patient (patient 5) presented with concurrent retinal vein occlusion in the left eye, which was resolved with successive intravitreal bevacizumab injections. Nine (45%) eyes of ten patients were hyperopic, and five (25%) eyes of three patients were diagnosed with angle-closure glaucoma.

All patients manifested bilateral multifocal vitelliform lesions on fundus photography with fovea-involving SRF and/or IRF on OCT B-scans (Appendix A). In addition, fundus autofluorescence imaging revealed marked hyperautofluorescence corresponding to yellow vitelliform lesions (Figure 1). The extent of vitelliform lesions was relatively consistent in all patients throughout the follow-up period, showing no correlation to BCVA. Subtle extramacular hyperautofluorescent deposits were often observed. Five (25%) eyes of three patients presented with peripheral drusen-like deposits, presumed to be located at the subretina [14], which appeared hypofluorescent on AF images (Figure 2).

OCT analysis revealed the presence of either SRF or IRF in both eyes of all the patients at the initial and final visits. Specifically, SRF was consistently sustained in all eyes during the follow-up period, whereas IRF was found in 11 (55%) eyes at the initial visit and in 9 (45%) eyes at the final visit. One patient (patient 8) received intravitreal bevacizumab injections due to misdiagnosis of chronic central serous chorioretinopathy, which was not effective in resolving the retinal fluid. One patient (patient 7) exhibited an increase in SRF and a newly developed IRF at the final visit without visual impairment. The extent or chronicity of the fluid did not correlate with BCVA. The CRT appeared to become progressively thinner, but there was no statistically significant difference between the initial and final values (Table 2). There was no significant change in the subfoveal CT during follow-up. Outer retinal layer thickening was found in 14 (70%) eyes of seven patients throughout the follow-up period, while the inner retinal layer remained relatively intact in the parafoveal areas.

One patient (patient 9) showed FCE in both eyes at the initial examination (Figure 3). Irregular PED (retinal pigment epithelial detachment) with subretinal hyperreflective materials was also found, indicating the possibility of an association with relatively indolent type 2 neovascular lesions [14].

All subjects harbored compound heterozygous mutations in the *BEST1* gene (Table 3). Nine unique disease-associated variants, including three novel mutations, have been reported. The most common mutation was c.584C→T, p.(Ala195Val), detected in six unrelated patients (patients 3, 5, 6, 7, 8, and 10). Patients 5 and 6 were siblings of the same pedigree, as previously reported [11]. Three missense mutations with uncertain significance included c.236C→T, p.(Ser79Phe), c.452C→T, p.(Leu151Pro), and c.650C→T, p.(Trp217Met).

Electrophysiological studies reflected impaired function of RPE and photoreceptor cells in both eyes almost symmetrically (Table 4). Twelve (60%) eyes of eight patients showed a prominent decrease or absence of a light peak, while the others showed subnormal Arden ratios in EOG. The ERG results were mostly within normal limits, except for two patients (patients 5 and 6). Both patients showed prominent thinning of the fovea and disruption of the inner segment/outer segment junction layer at the fovea, possibly due to chronic progressive destruction of macular function over 20 years.

## 4. Discussion

*BEST1* mutations have long been thought to act solely in an autosomal dominant manner. In 2008, Burgess et al. first defined autosomal recessive disease with *BEST1* mutation as a distinct category of bestrophinopathy, termed ARB [7]. Nearly 40 biallelic mutations in *BEST1* have been reported in patients with ARB to date [15,16,17]. Retinopathy in ARB includes irregularly distributed yellow deposits throughout the posterior fundus, which can be easily detected on autofluorescence imaging. Retinal edema and SRF are common findings in OCT imaging. ARB is associated with markedly abnormal EOG and relatively preserved ERG findings. Finally, the presence of compound heterozygous mutations in the *BEST1* gene is conclusive in the final diagnosis due to the diverse phenotypes of ARB and overlapping clinical features of bestrophinopathy spectrum diseases.

Burgess et al. speculated that ARB is the human null phenotype for the *BEST1* gene [10], which results from a complete loss of bestrophin-1 protein function within the RPE. Therefore, it was suggested that the autosomal recessive phenotype only manifests when bestrophin-1 activity drops below a functional threshold [10]. However, Li et al. suggested that partial loss-of-function mutations in the *BEST1* can also cause ARB, which presents less severe clinical features compared to null mutations [18]. As such, the initial presentation of the disease may be nonspecific or even asymptomatic due to incomplete penetrance. This may explain the lack of subjective symptoms and slow progressive decrease in central vision in our cohort. ARB has a wide age of onset, ranging from childhood to late adulthood. This trend can also be seen in our study, where disease onset ranged from the first to the sixth decade of life.

In this report, we analyzed the genetic and clinical characteristics of ten patients with ARB from nine unrelated families. Nine variants in *BEST1* were detected, including three novel mutations. Missense mutation was the only type of mutation detected in this cohort and it has been previously reported to be the most common type [19]. Despite being one of the most common retinal disorders caused by RPE mutations, a wide spectrum of bestrophinopathies are currently untreatable. Additionally, several cases of recessively inherited Best vitelliform macular dystrophy phenotype have been reported [20,21]. A simple dichotomy in inheritance patterns might lead to an inaccurate diagnosis of bestrophinopathies. Therefore, further studies are warranted to connect the functionality of the *B**EST**1* channel to the pathogenesis and progression of bestrophinopathies. Familial genetic studies in patients with novel mutations (c.236C→T in patient 2 and c.452C→T and c.650C→T in patient 4) might broaden our understanding of this disease entity.

Mutations in the *BEST1* gene cause abnormal functioning of bestrophin-1, Ca^2+^-activated Cl^−^ channels in the RPE, that are thought to generate EOG signals [6]. In electrophysiological studies, all patients showed abnormal EOG findings, suggesting a primary defect localized to the RPE. This explains the prevalent manifestation of SRF or IRF at the initial presentation due to dysfunction of the subfoveal or juxtafoveal RPE. However, central vision was relatively preserved as long as a certain amount of photoreceptor cells (outer retinal layer) were preserved, regardless of a profound amount of SRF and/or IRF at the fovea. Morphological OCT results indicated abnormal photoreceptor structure but unchanged inner retinal layers associated with abnormal cone responses in the central retina. However, bright flash ERG stimulates central and peripheral retina simultaneously, and ERG results indicate only global rod and cone photoreceptor function [15]. Therefore, the ERG results were mostly within normal limits, except for patients with profound disruption of photoreceptor cells or inner retina. The preservation of the inner retina and relatively normal ERG at an earlier stage demonstrate that early recognition and potential treatment at this stage might be a critical endpoint of prognosis in the future. 

Additionally, patients with betrophinopathies are at a risk of choroidal neovascularization. Large vitelliform lesions often hinder the detection of underlying neovascular membranes [22]. Several reports have pointed out favorable outcomes with the use of anti-VEGF therapy when choroidal neovascularization is present [22,23]. However, anti-VEGF injections for SRF/IRF in the absence of choroidal neovascularization appear to be ineffective in ARB. The distribution of SRF or IRF depends on the residual function of the RPE and not on the neovascular factors. In addition, the presence of SRF or IRF did not significantly interfere with the visual acuity. The use of anti-VEGF agents should be decided cautiously with the aid of fluorescence angiography, indocyanine green angiography, or OCT angiography. 

Our study has several limitations. The design of the study was retrospective and observational. Follow-up periods varied among patients which may have affected the analysis on clinical outcomes. Also, electrophysiological data was unavailable in some patients. Finally, only small number of subjects from a single ethnicity (Korean) were included in this study. Further studies on larger number of subjects from diverse ethnicity are warranted.

## 5. Conclusions

In conclusion, our study adds to the spectrum of clinical presentations caused by compound heterozygous mutations in the *BEST1* gene. Nevertheless, further studies are needed on the genetic analysis of the family of patients, as well as other patients with atypical forms of ARB. Multimodal imaging including wide-field fundus photography, autofluorescence, OCT, and EOG can be helpful in visualizing retinal abnormalities of clinically suspected ARB; however, genetic verification is recommended to confirm the diagnosis. Understanding the genetics and pathophysiology of bestrophinopathies may be useful to predict the prognosis and develop potential gene therapy in the future.

## Figures and Tables

**Figure 1 genes-13-01197-f001:**
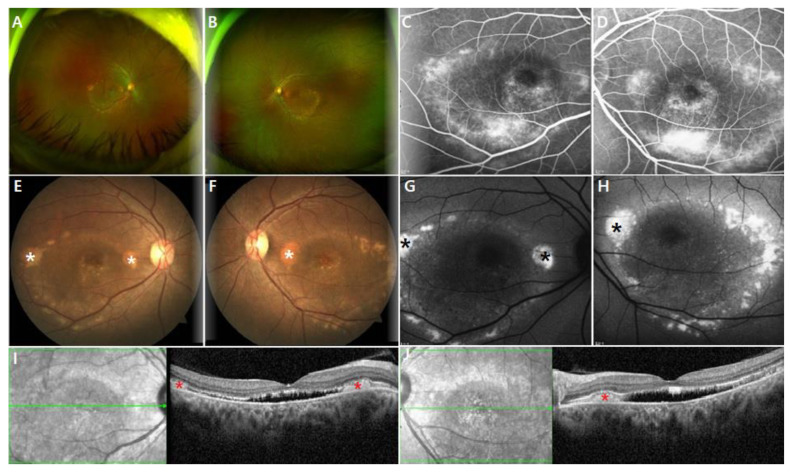
Multimodal retinal imaging of patient 7, a 37-year-old female with p.Ala195Val and p.Arg255Trp mutation in *BEST1*. At the initial visit, best corrected Snellen visual acuity was 0.5 for the right eye and 0.8 for the left eye, while the patient complained of subjective vision decrease only in the right eye despite the maculopathy in both eyes. (**A**,**B**) Widefield color images show bilateral multifocal vitelliform lesions. (**C**,**D**) Fluorescein angiography show non-specific hyperfluorescence. (**E**,**F**) Fundus photography clearly showing vitelliform lesions (white asterisks) in the posterior pole that topographically correspond to an increase in autofluorescence intensity (black asterisks) in 55° fundus autofluorescence images (**G**,**H**). (**I**,**J**) Spectral-domain optical coherence tomography B-scans also show subretinal hyperreflective deposits (red asterisks), subretinal fluid, and outer retinal layer thickening in both eyes.

**Figure 2 genes-13-01197-f002:**
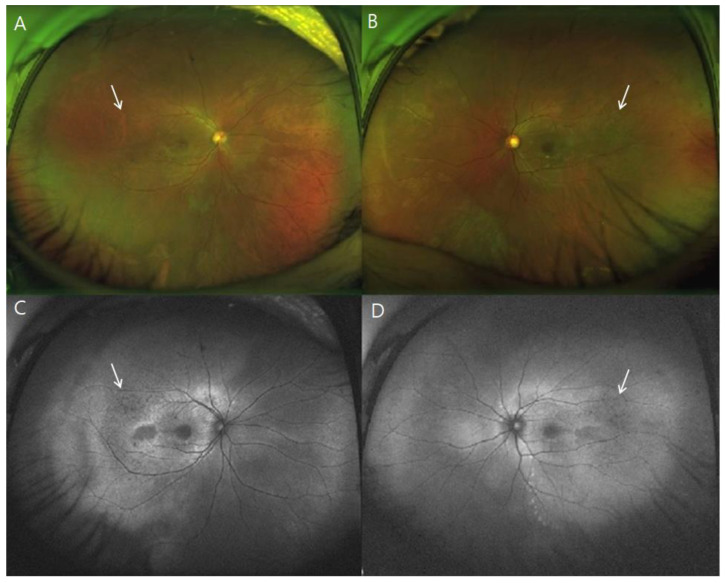
Widefield images of a 33-year-old man (patient 2, p.Arg255Trp and p.Ser79Phe mutation in *BEST1*). (**A**,**B**) Peripheral drusen-like materials (white arrows), presumed subretinal, are visible in widefield imaging, and appear hypofluorescent on widefield autofluorescence imaging (**C**,**D**). No evident peripheral retinal pigment epithelium atrophy was found.

**Figure 3 genes-13-01197-f003:**
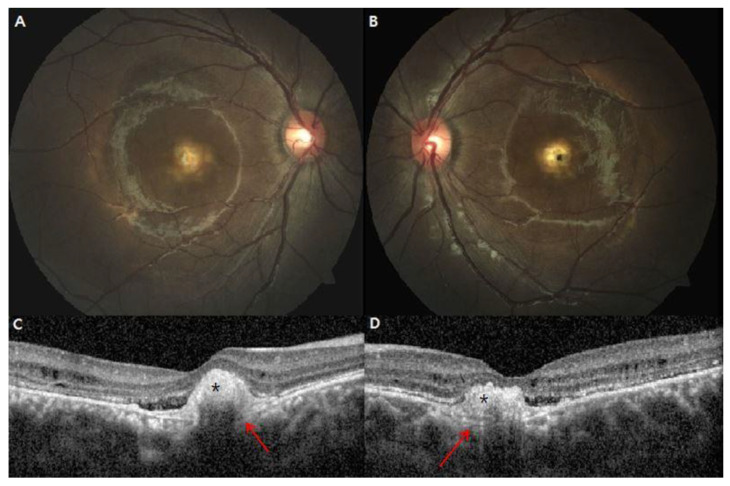
Focal choroidal excavation in both eyes of a 16-year-old male (patient 9, p.Arg47His and p.Ile38Ser mutation in *BEST1*) (**A**,**B**) Large juxtafoveal subretinal deposits with surrounding small deposits are present in both eyes. (**C**,**D**) The location of subretinal vitelliform material (black asterisks) topographically overlay the focal choroidal excavation (red arrows), challenging the detection of possible choroidal neovascularization at focal choroidal excavation.

**Table 1 genes-13-01197-t001:** Baseline characteristics.

Patient No.	Sex	Age at Onset (Year)	Age at Diagnosis (Year)	Follow-Up Period (Months)	Visual Acuity * at Initial Visit		Visual Acuity * at Final Visit		Spherical Equivalent		Primary Angle-Closure Glaucoma	Family History
					OD	OS	OD	OS	OD	OS		
1	F	33	35	52	0.15	0.5	0.1	0.4	−2.13	−1.25	Yes	Yes (brother)
2	M	30	33	11	1.0	0.7	1.0	0.7	−1.50	−3.50	Yes	No
3	M	10	16	14	1.0	0.6	1.0	0.5	1.50	1.88	No	No
4	F	32	32	44	0.7	1.0	0.5	0.5	−1.00	−0.88	Yes	No
5	F	20	53	105	0.025	0.025	0.04	0.04	4.25	4.00	No	Yes (brother)
6	M	25	67	145	0.1	0.1	0.04	0.04	2.75	2.63	No	Yes (Sister)
7	F	37	39	21	0.5	0.8	0.4	1.0	−0.63	−1.75	No	No
8	F	50	51	10	0.2	0.2	0.15	0.15	−0.50	0.13	No	Suspected (Sister)
9	M	15	16	12	0.3	0.5	0.3	0.5	−1.00	−0.50	No	No
10	F	6	6	14	0.7	0.8	0.4	0.4	2.63	3.13	No	No

Abbreviations: OD, right eye; OS, left eye; F, female; M, male. * Snellen visual acuity.

**Table 2 genes-13-01197-t002:** Spectral-domain optical coherence tomography parameters of 20 eyes with autosomal recessive bestrophinopathy at initial and final visits.

OCT Parameters	At Initial Visit	At Final Visit
Macular subretinal deposit (%)		
Unifocal	30	25
Multifocal	60	60
Subretinal fluid (%)		
Sub-foveal	70	60
Diffuse	10	20
Intraretinal fluid (%)	50	40
Focal choroidal excavation (%)	10	10
Central macular thickness (μm) *	198.8 ± 231.2/173.7 ± 167.4	185.9 ± 196.3/153.9 ± 150.7
Subfoveal choroidal thickness (μm) *	303.4 ± 70.5/280.4 ± 76.7	289.3 ± 61.9/286.5 ± 47.4
Outer retinal layer thickening (%)	70	70

Abbreviations: OCT, optical coherence tomography. * Presented as right eye/left eye. Data are shown as mean ± standard deviation.

**Table 3 genes-13-01197-t003:** List of *BEST1* mutation variants and predicted effects.

Patient No.	*BEST1* Mutation	Amino Acid Change	*BEST1* Mutation	Amino Acid Change
1	c.763C>T	p.Arg255Trp	c.113T>G	p.Ile38Ser
2	c.763C>T	p.Arg255Trp	c.236C>T	p.Ser79Phe
3	c.584C>T	p.Ala195Val	c.763C>T	p.Arg255Trp
4	c.452C>T	p.Leu151Pro	c.650C>T	p.Trp217Met
5	c.119T>C	p.Leu40Pro	c.584C>T	p.Ala195Val
6	c.119T>C	p.Leu40Pro	c.584C>T	p.Ala195Val
7	c.584C>T	p.Ala195Val	c.763C>T	p.Arg255Trp
8	c.584C>T	p.Ala195Val	c.763C>T	p.Arg255Trp
9	c.140G>A	p.Arg47His	c.113T>G	p.Ile38Ser
10	c.584C>T	p.Ala195Val	c.632T>C	p.Leu211Pro

**Table 4 genes-13-01197-t004:** Electrophysiologic study of patients.

Patient No.	ERG		EOG (Arden Ratio)	
	OD	OS	OD	OS
1	WNL	WNL	1.1	1.8
2	WNL	WNL	1.9	1.4
3	WNL	WNL	1.8	1.5
4	WNL	WNL	1	1
5	Rod and cone impairment		1.5	2
6	Rod and cone impairment		0.92	1.06
7	Not performed		Not performed	
8	Not performed		Not performed	
9	WNL	WNL	1.1	1
10	Not performed		1.5	1.1

Abbreviations: ERG, electroretinogram; EOG, electrooculogram; OD, right eye; OS, left eye; WNL, within normal limits.

## Data Availability

Data presented in this study is contained within the article.

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
