# Peer review of "Clinical Features and Genetic Findings of Autosomal Recessive Bestrophinopathy"

_genes, 2022, doi:10.3390/genes13071197_

Round 1
Reviewer 1 Report
In this short clinical report, the authors provide an overview of the phenotypes associated with Autosomal Recessive Bestrophinopathy (ARB), share clinical imaging data, and report SNPs in BEST1 that appear to be causative of retinal disease. This is a retrospective study from two hospitals; all cases reported are compound heterozygotes and represent male and female-identifying patients across a wide range of ages (16-80). Sequencing from blood samples identified 6 known variants in BEST1 and 3 novel missense mutations. This is the largest number of ARB cases reported in one paper and emphasizes the similarities in phenotype between ARB and known dominant (and/or haploinsufficient) forms of Bestrophinopathy.
Overall the data appear sound, and from a purely scientific perspective this seems like an important study, but I have major concerns about the ethics of no informed consent. From the methods it is difficult to tell if blood samples were taken after the ophthalmological exam, which seems like a point in time that informed consent could have been solicited. A clinician/clinical study expert needs to review this paper.
A few minor questions/concerns:
1) Need more clarity around why the particular images were chosen for the main paper and explicit references to and description of the supplementary data within the Results section.
2) More extensive figure captions to describe what is shown. For example, there are arrows on Figure 2 but what they are pointing to is not described in the caption. Moreover, the asterisks in Figure 1 are very small and difficult to see.
3) Need to say at the beginning of the results that all examples found were compound heterozygotes instead of waiting for the discussion.
4) Is there a correlation between severity of vision loss and other phenotypes such as extent of vitelliform lesions and subretinal fluid?
5) What exactly does asymptomatic on line 124 mean in the context of this paper? No vision loss? Was the asymptomatic patient always asymptomatic?
6) What exactly does “null mutation” on line 179 mean in the context of this paper. I tend to think of “null” as no expression of the gene and/or a premature STOP codon. Need to be clear how “null” was determined.
7) A short discussion of why there might be a difference in ERG but not EOG in ARB patients would helpful for the non-clinician.
Author Response
We appreciate the thoughtful and favorable comments of reviewer on our study. We added additional explanations to the comments you gave, which are described in detail below.
Point1. Overall the data appear sound, and from a purely scientific perspective this seems like an important study, but I have major concerns about the ethics of no informed consent. From the methods it is difficult to tell if blood samples were taken after the ophthalmological exam, which seems like a point in time that informed consent could have been solicited. A clinician/clinical study expert needs to review this paper.
Response1. As the editorial board requested the acquisition of consent from the all patients, written informed consent has been obtained from the patients for the publication of this paper. Also, we added the statement of informed consent for genetic analysis after ophthalmological exam on line 110-111.
A few minor questions/concerns:
1) Need more clarity around why the particular images were chosen for the main paper and explicit references to and description of the supplementary data within the Results section.
→ We added the description of the supplementary data on line 139. Figure 1 was chosen for the main paper because patient 7 presented most prominent multiple vitelliform lesions at the initial visit and this would help easier understanding of other researchers.
2) More extensive figure captions to describe what is shown. For example, there are arrows on Figure 2 but what they are pointing to is not described in the caption. Moreover, the asterisks in Figure 1 are very small and difficult to see.
→ We added the explanation for white arrows in caption of Figure 2. Also, we increased the size of asterisks in Figure 1 for better visualization.
3) Need to say at the beginning of the results that all examples found were compound heterozygotes instead of waiting for the discussion.
→ On line 190, we stated that all examples were compound heterozygous mutations in BEST1 gene.
4) Is there a correlation between severity of vision loss and other phenotypes such as extent of vitelliform lesions and subretinal fluid?
→ There was no correlation between vision loss and the extent of vitelliform lesions or subretinal fluid. We added sentence “The extent of vitelliform lesions was relatively consistent in all patients throughout the follow-up period, showing no correlation to BCVA.” on line 141-143. Meanwhile, on line 169-170, we stated that “The extent of fluid or chronicity of fluid did not correlate with visual acuity” indicating irrelevant correlation. The sentence on line 246-249, “However, central vision was relatively preserved as long as a certain amount of photoreceptor cells (outer retinal layer) was preserved, regardless of a profound amount of SRF and/or IRF at the fovea.”, also adds to our claims.
5) What exactly does asymptomatic on line 124 mean in the context of this paper? No vision loss? Was the asymptomatic patient always asymptomatic?
→ Two patients were diagnosed incidentally on medical check-ups and had no subjective symptom at all. To clarify the subjective symptom change, we added sentence “At the final visit, regardless of change in BCVA, three patients (patients 6, 8, and 10) experienced gradual vision decrease while the others claimed no vision change.” on line 129-131.
6) What exactly does “null mutation” on line 179 mean in the context of this paper. I tend to think of “null” as no expression of the gene and/or a premature STOP codon. Need to be clear how “null” was determined.
→ We changed “null” to “novel” mutation to clarify the meaning of newly found mutation in BEST1 gene of autosomal recessive bestrophinopathy patients.
7) A short discussion of why there might be a difference in ERG but not EOG in ARB patients would helpful for the non-clinician.
→ We added explanation of ERG and EOG results on line 242-254 in discussion for better understanding of non-clinicians.
Reviewer 2 Report
In this manuscript, the authors reported the clinical phenotypes and genetic diagnosis of 10 autosomal recessive bestrophinopathy (ARB) patients from nine unrelated families, and identified three novel disease-causing mutations of the BEST1 gene, namely c.236C→T, p.(Ser79Phe), C.452C→T, p.(Leu151Pro), and c.650C→T, p.(Trp217Met). Overall, the results are solid and reveal novel BEST1 mutations associated with ARB.
That being said, the following points need to be addressed.
Major concern: the author need to better integrate and cite recent findings in the literature regarding the function of Best1.
1. Line 34, “The exact function of the Best1 is not completely understood.” This is not informative. The authors need to describe the channel function of Best1 and cite the appropriate references.
2. Line 35-36, “However, it has been suggested that mutation in the BEST1 gene might result in abnormal function of anion channel in RPE…”. References need to be cited here.
3. Line 50-51, “ARB is caused by compound heterozygous BEST1 mutation with a modifier effect of the first onto the second mutation.” This claim is not true. Some ARB is caused by homozygous mutation of BEST1.
4. Line 206-209, “Burgess et al. speculated ARB as the human null phenotype for BEST1… it is suggested that the autosomal recessive phenotype only manifests when the bestrophin-1 activity drops below a functional threshold.” This claim was made in the original paper connecting ARB to BEST1 recessive mutations published 15 years ago. Recent studies strongly indicate it is not the case. For instance, I201T is a partial loss-of-function mutation (thus not null) that causes ARB, while many BVMD-associated BEST1 dominant mutations are also loss-of-function.
Minor points:
1. Line 13, “recently…” seems a stretch here as the discovery was made in 2008.
2. Line 170, please explain “PED”.
3. Line 179, “null” should be “new (or novel)”.
4. Line 218-220, “A wide spectrum of bestrophinopathies is currently untreatable hereditary disease despite being one of the most common retinal disorders caused by RPE mutation.”. This sentence is awkward and needs to be reframed.
5. Line 222-224, “further studies are warranted to clarify the function of BEST1 gene in pathogenesis and progression of bestrophinopathies”. Probably revise to “connect the functionality of Best1 channel to pathogenesis and progression of bestrophinopathies”.
Author Response
We appreciate the thoughtful and favorable comments of reviewer on our study. We added additional explanations to the comments you gave, which are described in detail below.
Major concern: the authors need to better integrate and cite recent findings in the literature regarding the function of Best1.
- Line 34, “The exact function of the Best1 is not completely understood.” This is not informative. The authors need to describe the channel function of Best1 and cite the appropriate references.
→ We deleted the sentence “The exact function of the Best1 is not completely understood.” Also we added reference 5 [Yang T.; Justus S.; Li Y.; BEST1: the Best Target for Gene and Cell Therapies. Molecular Therapy. 2015, 23, 1805–1809] and reference 6 [Xiao Q.; Hartzell C.; Yu K.; Bestrophins and Retinopathies. Pflugers Arch. 2010, 460, 559–569] to describe the ion channel function of BEST1 gene.
- Line 35-36, “However, it has been suggested that mutation in the BEST1 gene might result in abnormal function of anion channel in RPE…”. References need to be cited here.
→ We added reference 5 [Yang T.; Justus S.; Li Y.; BEST1: the Best Target for Gene and Cell Therapies. Molecular Therapy. 2015, 23, 1805–1809] and reference 6 [Xiao Q.; Hartzell C.; Yu K.; Bestrophins and Retinopathies. Pflugers Arch. 2010, 460, 559–569] on line 37.
- Line 50-51, “ARB is caused by compound heterozygous BEST1 mutation with a modifier effect of the first onto the second mutation.” This claim is not true. Some ARB is caused by homozygous mutation of BEST1.
→ Considering the reference 9 and reference 10, we added the comment of homozygous mutation of BEST1 in ARB on line 50-52 as well as on line 14 of abstract.
- Line 206-209, “Burgess et al. speculated ARB as the human null phenotype for BEST1… it is suggested that the autosomal recessive phenotype only manifests when the bestrophin-1 activity drops below a functional threshold.” This claim was made in the original paper connecting ARB to BEST1recessive mutations published 15 years ago. Recent studies strongly indicate it is not the case. For instance, I201T is a partial loss-of-function mutation (thus not null) that causes ARB, while many BVMD-associated BEST1dominant mutations are also loss-of-function.
→ On line 222-224, citation of partial loss-of-function mutation in the BEST1 gene was added. As a result of diversity in BEST1 mutation variants, the initial presentation of ARB may vary from milder form to severe impairment of the central vision.
Minor points:
- Line 13, “recently…” seems a stretch here as the discovery was made in 2008.
→ We deleted “recently recognized” on line 13.
- Line 170, please explain “PED”.
→ We added “(retinal pigment epithelial detachment)” to explain PED on line 181.
- Line 179, “null” should be “new (or novel)”.
→ We changed “null” to “novel” on line 191.
- Line 218-220, “A wide spectrum of bestrophinopathies is currently untreatable hereditary disease despite being one of the most common retinal disorders caused by RPE mutation.”. This sentence is awkward and needs to be reframed.
→ We reframed the sentence on line 233-235 as “Despite being one of the most common retinal disorders caused by RPE mutations, a wide spectrum of bestrophinopathies is currently untreatable.”
5. Line 222-224, “further studies are warranted to clarify the function of BEST1 gene in pathogenesis and progression of bestrophinopathies”. Probably revise to “connect the functionality of Best1 channel to pathogenesis and progression of bestrophinopathies”.
→ We revised the sentence on line 238-239 as you recommended.